**METHOD**

# ISLET: individual-specific reference panel recovery improves cell-type-specific inference

Hao Feng[1*], Guanqun Meng[1], Tong Lin[2], Hemang Parikh[3], Yue Pan[2], Ziyi Li[4], Jeffrey Krischer[3] and Qian Li[2*]

*Correspondence:
hxf155@case.edu; qian.li@stjude.org

[1] Department of Population and Quantitative Health Sciences, Case Western Reserve University, Cleveland, OH, USA
[2] Department of Biostatistics, St. Jude Children's Research Hospital, 262 Danny Thomas Place, Memphis, TN 38105, USA
[3] Health Informatics Institute, University of South Florida, Tampa, FL 33620, USA
[4] Department of Biostatistics, The University of Texas MD Anderson Cancer Center, Houston, TX 77030, USA

## Abstract

We propose a statistical framework ISLET to infer individual-specific and cell-type-specific transcriptome reference panels. ISLET models the repeatedly measured bulk gene expression data, to optimize the usage of shared information within each subject. ISLET is the first available method to achieve individual-specific reference estimation in repeated samples. Using simulation studies, we show outstanding performance of ISLET in the reference estimation and downstream cell-type-specific differentially expressed genes testing. We apply ISLET to longitudinal transcriptomes profiled from blood samples in a large observational study of young children and confirm the cell-type-specific gene signatures for pancreatic islet autoantibody. ISLET is available at https://bioconductor.org/packages/ISLET.

**Keywords:** Deconvolution, Temporal measures, Cell-type-specific differential expression, Individual-specific reference panel

## Background

Clinical samples often contain a mixture of different cellular subpopulations. The real-world clinical RNA-seq signatures are, therefore, mosaics of signals from multiple pure cell types. As a result, the observed bulk RNA-seq data can be viewed as the weighted average of signals from multiple cell types, whereas the weights, naturally, are the proportions in the mixture. Over the last decade, researchers have gained substantial interests in computational methods to deconvolute cell population frequencies. For example, CIBERSORT [1], CIBERSORTx [2], TIMER [3], MuSiC [4], and DWLS [5] were proposed to estimate cellular composition and infer cell-type-specific expression profiles. These new methods improved the resolution of traditional analytical approaches to identify bulk RNA-seq <u>D</u>ifferentially <u>E</u>xpressed <u>G</u>enes (DEG).

 Recently, novel methods including CARSeq [6], TOAST [7, 8], and TCA [9] were developed to incorporate pre-estimated cell mixture proportions in the modeling of observed bulk data and directly detect DEG at cell-type-specific resolution. This new type of DEG analysis at cell type level is generally referred to as cell-type-specific

Differentially Expressed Genes identification. The rapid methodological advancement in cell proportion deconvolution and csDEG calling during the recent years has offered refined tools for cell-type-aware knowledge mining. It is worth noting that although single-cell RNA-seq (scRNA-seq) or cell sorting from samples can directly measure signals from each cellular subpopulation, the costs and technical challenges in cell isolation is still a major hurdle to apply scRNA-seq in population-based study. For example, to investigate immunity-related signals in the blood samples in a multi-center cohort study, sorting peripheral blood mononuclear cells (PBMC) and profiling transcriptomes is cost prohibitive [10, 11]. Hence, bulk RNA-seq profiling is still favorable in medium- or large-scale studies.

The longitudinal or repeated sampling has become a popular strategy in clinical transcriptomic research, such as a longitudinal bulk RNA-seq study coupled with small-scale scRNA-seq profiling in high grade serous ovarian cancer [12, 13]. Another recent study [14] using longitudinal whole blood bulk gene expression found that the gene markers contributing to the development of type 1 diabetes (T1D) were also related to blood immune cells. To discover the latent dynamics within each cellular subpopulation without gene network construction, one should develop a novel framework that directly identifies temporal cell-type-specific gene signatures associated with disease status.

Despite numerous methods developed for bulk data deconvolution and csDE analysis [4, 7, 15–18], limitations exist due to the rare usage of participant indicator and lack of participant-level reference profile. Traditionally, these methods were developed under the assumption of one identical feature-by-cell type reference panel across all the samples, which were not capable of characterizing individual cell-type-specific gene expression reference. In contrast, the algorithms in recent tools TCA [9] recovered the reference panel for each sample without specifying the between-subject heterogeneity [19]. Another concern in these methods is the strong heterogeneity within each subject, as time-dependent variation in transcriptomes may not be profound in subgroups of participants [20]. Hence, a subject-specific reference information shared by different time points is more reasonable than assuming independent reference panels across time points. To accurately detect cell-type-specific gene signatures, an individual-specific reference panel should be reconstructed for longitudinal bulk gene expression data.

Here, we present a novel computational method ISLET (Individual Specific celL typE referencing Tool), to estimate the cell-type-specific gene expression reference panel for each participant. The unobserved panel per subject are estimated by the expectation-maximization (EM) algorithm in a mixed-effect regression model. ISLET leverages multiple or temporal observations of each subject, to construct a likelihood-based statistics for csDEG inference. This is the first statistical framework to recover the subject-level reference panel by employing multiple samples per subject. Our model also provides the flexibility to incorporate group-wise change rate across time points in the reference gene expression and then identify differential dynamics in each cellular subpopulation.

We designed an extensive simulation study to compare our method ISLET with csSAM, TOAST, TCA, CARSeq, aDESeq2 [6, 7, 9, 15, 21], demonstrating ISLET being a powerful and robust tool that outperformed these competing methods. We applied ISLET to the longitudinal bulk RNA-seq data in The Environmental Determinants of Diabetes in the Young (TEDDY) cohort [14] and the Parkinson's Disease Biomarker Program (PDBP). Our

method successfully identified potential gene signatures differentially expressed in B cells, natural killer (NK) cells prior to the onset of pancreatic $\beta$-cell autoantibodies, and those in CD8+ T cells, NK cells for Parkinson's disease. ISLET is available on Bioconductor at https://bioconductor.org/packages/ISLET [22]. Overall, the improved resolution of subject-specific and cell-type-specific biomarkers, estimated by ISLET, helped to achieve more precise biomarker associations with clinical outcomes.

## Results

### Deconvolution of individual reference profile and temporal csDE analysis

Our framework utilizes repeatedly measured bulk transcriptomes as input to retrieve a cell-type-specific gene expression profile for each participant (subject) and perform the csDE analysis, as illustrated in Fig. 1. Besides the bulk transcriptomic data, it also takes cell type proportions, subject disease status, and optional covariates as the input. Note that the cell type proportions can be estimated by an existing tool (such as CIBERSORT, MuSiC, TOAST, or AutoGeneS [1, 4, 7, 23]) in practice and thus is treated as a known input in our model. As an example, each subject or participant $j$ has whole blood or PBMC samples collected at multiple time points $t = 1, \ldots, T_j$. The gene expression measured in participant $j$'s sample at time point $t$ is $y_{jt}$, and $z_j$ is the indicator for each individual's treatment group, phenotype, or case-control status. The gene expression value $y_{jt}$ is either raw or normalized counts without batch effect and can be replaced by DNA methylation measurement if the feature is a CpG site. There are $K$ cell types of interest in each sample, with estimated or known proportions $\theta_{jtk}$, and naturally $\sum_{k=1}^{K} \theta_{jtk} = 1$. Given the input cell proportions, the observed bulk gene expression $y_{jt}$ can be described by a linear mixed effect model

$$E(y_{jt}) = \sum_{k=1}^{K} \left( m_k + \beta_k z_j + u_{jk} \right) \theta_{jtk} \tag{1}$$

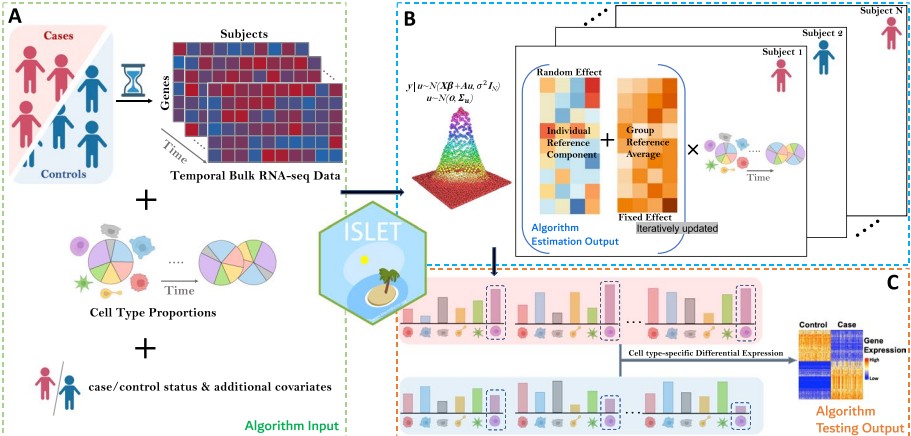

**Fig. 1** An overview of our proposed method ISLET (Individual Specific celL typE referencing Tool). **A** ISLET takes repeatedly measured bulk RNA-seq data, cell type proportions (known or estimated), and disease status as the algorithm input. Additional covariates are optional. **B** By a hierarchical mixed-effect modeling, ISLET can iteratively retrieve individual-specific and cell-type-specific gene expression reference panels. The fixed effect is the group-level average and the random effect is the individual-level deviance from the group mean. **C** Given the individual-specific reference panel, ISLET can conduct test to identify cell-type-specific differentially expressed genes (csDEG)

where $m_k$ is the mean gene expression in cell type $k$ for the baseline group (i.e., controls, $z_j = 0$), and $m_k + \beta_k$ is the mean expression in cell type $k$ for cases ($z_j = 1$). The random variable $u_{jk}$, shared by the repeated or longitudinal samples from the same subject, represents a subject-specific deviation in the group-wise mean expression in cell type $k$. This unobserved random term accounts for the within-subject measurements correlation. Hence, we can deconvolve the mixture-cell gene expression into $K$ cell types for each subject, individually, by estimating the fixed and random effects ($m_k, \beta_k, u_{jk}$), as illustrated in Fig. 1. To detect csDEG between cases and controls, one can test the hypothesis $H_0 : \beta_k = 0$.

This framework can be extended by adding time-related change in the reference panels, with change rate fixed by group. For simplicity, we assume linear change rate (i.e., slope) in the present research. The model in (1) can be generalized as

$$E(y_{jt}) = \sum_{k=1}^{K} \left( \tilde{m}_k + \tilde{\alpha}_k c_{jt} + \tilde{\beta}_k z_j + \gamma_k c_{jt} z_j + u_{jk} \right) \theta_{jtk} \tag{2}$$

with $c_{jt}$ denoting the time or sample age and $\tilde{\alpha}_k$ denoting the slope in baseline group. The other potential covariates can be included in $c_{jt}$ as a vector. The parameter $\tilde{\beta}_k$ represents age-independent difference between groups in cell-type-specific gene expression. A differential slope $\gamma_k \neq 0$ implies that the cases' gene expression change over time in cell type $k$ is associated with the disease status $z_j$. There may be no significant group effect in the intercept ($\tilde{\beta}_k = 0$) or slope ($\gamma_k = 0$), but the participants still have distinct underlying reference profiles ($u_{jk}$). The csDEG in intercept or slope can be identified by testing $H_0 : \tilde{\beta}_k = 0$ or $H_0 : \gamma_k = 0$, respectively. Parameters estimation from the modeling above can be achieved by adopting expectation-maximization (EM) algorithm and is detailed in the "Methods" section.

### Simulation study

#### *Temporal cell type proportions*

The cell type proportions were generated based on real single-cell RNA-seq (scRNA-seq) datasets, from annotated and well-labeled cell types. Here, we compiled a pool of cell type labels by aggregating multiple scRNA-seq datasets cell labels and conducted bootstrap to generate cell type labels and calculate the cell type proportions, from each resampling. Those bootstrapped proportions were then fitted by the Dirichlet distribution to estimate the parameters $\alpha$. Note that the number of cell types can be customized during this procedure. This enables us to simulate cell type compositions in different scenarios. For example, a study could have one dominant cell type and several minor cell types; or in contrast, a study could have relatively balanced cell type proportions with low composition variation. This procedure allows us to obtain cell type proportions that best mimic real data. Details of cell type proportion simulation are described in "Methods" section.

#### *Individual reference panel and temporal gene expression*

Cell type-specific underlying expression reference profile were also generated based on real cell line RNA-seq dataset. All parameters were derived from a real study (GSE60424) and solved by the Bioconductor package PROPER [24]. This real dataset

collected immune-associated diseases and pure immune cell line bulk RNA-seq data. The expression mean and biological dispersions per gene and cell type are estimated by *estParam* function in PROPER at log scale. This real data suggests strong associations across cell types in mean expression and biological dispersion. Thus, we adopted multivariate normal (MVN) distribution to compute the variance-covariance information, $\mathbf{\Sigma_m}$ and $\mathbf{\Sigma_\phi}$, respectively, for mean and overdispersion parameters. For gene $g$, the mean expression and overdispersion per subject $j$ are simulated by MVN with $\bar{\mu}_m$ and $\bar{\mu}_\phi$, who are K-dimensional vectors representing the mean and overdispersion parameters estimated from PROPER.

To simulate the biological variation, we adopted a Gamma distribution to generate the true cell-type-specific gene expression matrix. We assume cases and controls have identical gene-wise overdispersion $\mathbf{\Sigma_\phi}$ but different mean expression specified by log fold change (LFC). The reference gene expression per control (baseline group) are simulated by the above process with mean $\bar{\mu}_m^{ctr}$, while the expression for cases are generated by the same process with mean $\bar{\mu}_m^{case} = \bar{\mu}_m^{ctr} + \Delta$, where $\Delta$ is LFC. The true mixture-cell expression per gene per sample ($\bar{\lambda}_{gjt}$) is the weighted sum of the reference gene expression, with weights as cell proportions. We set 10% genes as DE in a cell type, although two of the six cell types do not contain any DE genes. The values of LFC are set at $\mathrm{LFC} = 0, 0.5, 0.75, 1.0, 1.25, 1.5$. The number of subjects is set at $N = 50, 100, 150, 200$. Each subject has measurements at three time points, and the number of subjects are equal between two groups. The observed raw counts is generated by Poisson distribution with mean $\bar{\lambda}_{gjt}$ to mimic the technical noise in sequencing experiments.

### Reference panel deconvolution

We first evaluated the accuracy of ISLET on individual-specific reference panel recovery, using the synthetic data outlined above. Since ISLET is the only method that estimates subject-specific reference expression from longitudinal bulk samples, the comparison with existing methods in subject-specific reference panel is not directly viable. Nevertheless, we chose TOAST and TCA for a detailed comparison. TOAST can solve for one reference panel per group (case/control), and this group-wise reference matrix is then treated as the subject-specific reference panel in evaluation. TCA can solve for sample-specific reference panel; thus, we use the average of multiple estimated reference panels per subject to obtain the subject-specific profile. These two approaches reflect the two ends of the deconvolution spectrum: a fixed reference panel across subjects (or samples) per group vs. a sample-specific reference, while ISLET fills the methodology gap in between.

For the aforementioned three methods, cell type proportions were provided as input. We assessed the reference estimation accuracy in two scenarios, in which the true and pre-deconvoluted proportions served as the input, respectively. The pre-deconvoluted proportions were obtained by CIBERSORTx using noise-added reference panels, with 200 marker genes identified independently by TOAST package based on coefficient of variations. There are $N = 50$ subjects per group, each has three samples under the null hypothesis (0% DEG). The ISLET-estimated individual reference panel, based on the true or pre-deconvoluted cell type composition, are shown in scatterplots (Fig. 2A, B). The true cellular proportions (Fig. 2A) yields more precision in reference panel estimation,

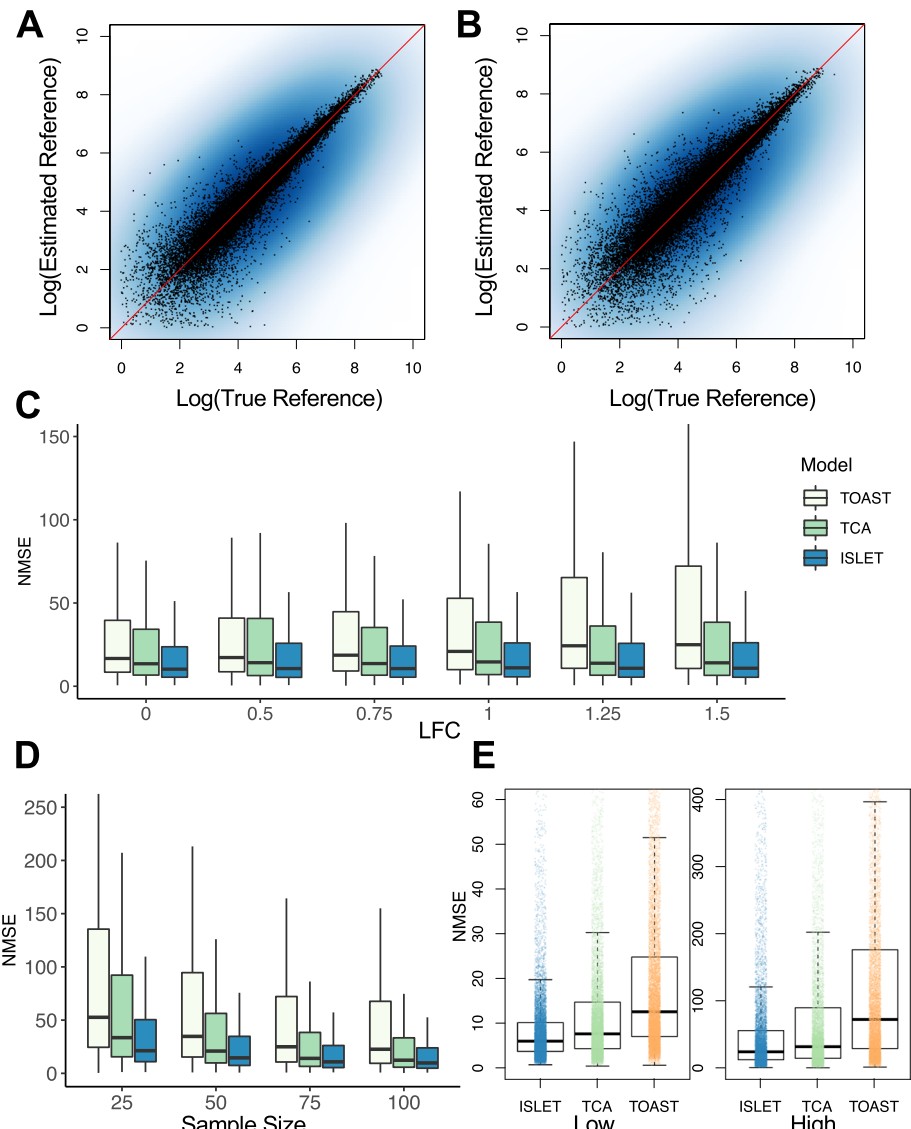

**Fig. 2** ISLET accurately estimates individual-specific gene expression reference panels. **A** Scatterplot showing ISLET estimated reference panel versus the true reference panel, in log scale, when using the true cell type proportions as the input. **B** Scatterplot similar to panel **A** but using the estimated cell type proportions as the input. **C** Normalized mean squared error (NMSE) in reference panel estimation, from three methods: TOAST, TCA, and ISLET, at various effect sizes. Log fold change (LFC) used in simulation: 0, 0.5, 0.75, 1, 1.25, and 1.5. **D** NMSE at various samples sizes per group (25, 50, 75, and 100), comparing three methods. **E** NMSE stratified by gene expression level (low expression: ≤160 and high expression: > 160). $N = 20$ simulations are conducted in each scenario

compared with the computationally deconvoled proportions (Fig. 2B). This difference is more profound in low-expression genes (log-expression < 7).

We next assessed if the effect size of DEG could have an impact on reference estimation. The effect size is reflected through the log-fold change (LFC) value of the synthetic DEG. Figure 2C shows the estimation error, represented by normalized mean squared error (NMSE) for the three methods across various choices of LFC ranging from 0 (under the null) to 1.5 (large effect size). ISLET has the lowest median and lowest

variation NMSE under all choices of effect sizes, showing the most accurate and stable estimation. In addition, ISLET is robust with the least NMSE increase at larger effect sizes. In contrast, TOAST have inflated error at large LFC and TCA has higher variability. The estimation accuracy at different sample sizes is shown in Fig. 2D. As expected, all methods benefited from an increased sample size. ISLET achieved the lowest error consistently at sample sizes of 25 to 100 per group. It has the lowest median error as well as the lowest variation. Next, to investigate if the expression level would impact the estimation accuracy, we stratified the genes into two groups: low expression genes (expression count value ≤160) and high expression genes (expression count value > 160). As shown in Fig. 2E, the bias of ISLET is the lowest at both strata, showing the most robust performance among the three methods. Similar conclusions can be drawn in other combinations of LFC and sample size (Additional file 1: Figs. S1-S7 ). Side-by-side comparisons using ground true proportions versus estimated proportions (Fig. S2) indicates that imprecise cell type proportions would impact reference panel estimation negatively. Our method ISLET still achieves the lowest NMSE on average and the smallest variation in each simulation scenario, compared with other methods. These results, overall, highlight both the unfavorable impact of imprecise proportions to reference panel estimation, and the merits of our modeling approach compared to others. Furthermore, our method still outperforms TOAST if the TOAST deconvolution is performed on each subject's repeated samples, individually.

### Improved csDEG identification

With the improved reference estimation demonstrated above, we further investigated ISLET performance in cell-type-specific differentially expressed genes (csDEG) identification by using the model in Equation (1). We compared ISLET with five methods, i.e., CARseq, TOAST, TCA, DESeq2, and csSAM. All the competing methods used the true cell type proportions along with bulk gene expression values as input. The csDE analysis in DESeq2 can be performed by testing the coefficients for interaction terms of each cell type and group.

The result is shown in Fig. 3 and benchmarked by metrics of true discovery rate (TDR), receiver operating characteristic (ROC), and sensitivity and false discovery rate (FDR). Here, the TDR is defined as the proportion of true csDEG, among the top-ranking identified genes for each method. TDR is equivalent to precision and reflects a practical consideration: biologists may focus on the top-ranking significant csDEG output given a certain method; thus, the accuracy among them matters the most for biomarker discovery. Figure 3A shows the averaged results for 20 replicates, among all six methods, for the sample size of $N = 25$ subjects per group at one cell type. ISLET shows the highest precision among top-ranking genes in TDR and the largest AUC in ROC. We also examined the distributions of sensitivity and FDR, for each method, among all cell types. The third panel of Fig. 3A shows joint distribution of sensitivity and FDR, given the intended FDR level 0.1. Apparently, ISLET yields the highest sensitivity and reasonably controlled FDR. TCA can outperform ISLET in terms of sensitivity only in very few datasets, but its seriously inflated FDR is a concern. DESeq2 also resulted in poor FDR, due to its inflated type I error for empirical Bayes approach under large sample sizes [25]. The sensitivity plot in the fourth column of Fig. 3A demonstrates that ISLET is the most powerful

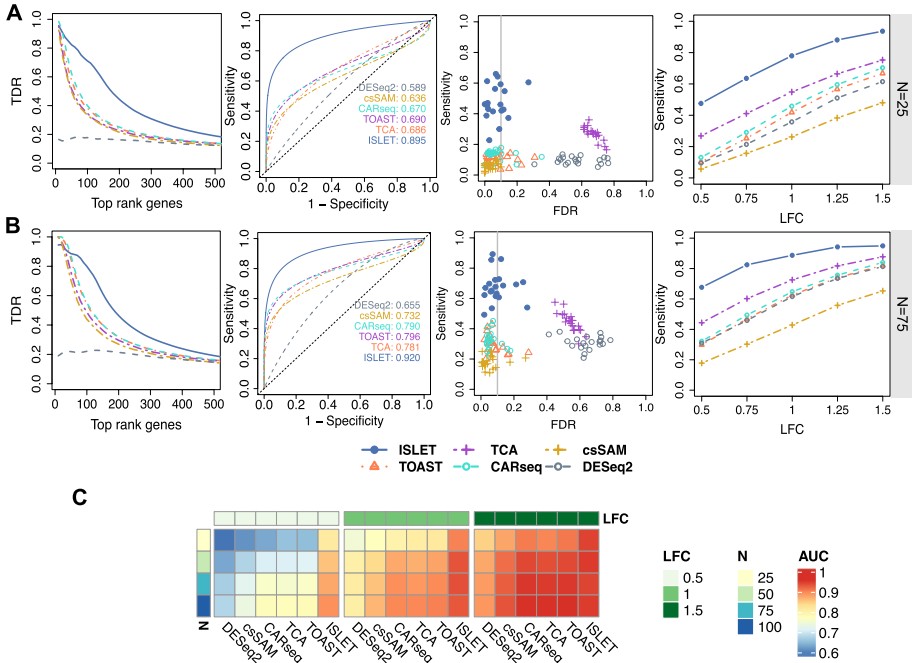

**Fig. 3** ISLET improves testing accuracy in cell-type-specific differentially expressed genes (csDEG) identification. **A** True discovery rate (TDR), receiver operating characteristic (ROC), sensitivity versus false discovery rate (FDR), and sensitivities at various effect sizes are shown from left to right. Results for one cell type, at sample size of $N = 25$ subjects per group with log fold change (LFC) = 0.5, are shown for the first three panels. Intended FDR level of 0.1 (vertical line) in the third panel. The fourth panel shows the averaged sensitivity across all cell types, at various LFCs. Benchmarked methods include CARseq, TOAST, TCA, DESeq2, and csSAM. **B** Displaying the same metrics as in **A** except sample size per group is $N = 75$. **C** Averaged AUC values across all cell types, at various combinations of sample sizes and LFCs. $N = 20$ simulations are conducted for each simulation scenario described above

method for various LFC with $N = 25$ subjects per group. All the competing methods show increased sensitivity at larger LFC.

Simulations at a larger sample size of $N = 75$ per group show a similar conclusion as the above, in terms of the same metrics (Fig. 3B). In addition, similar conclusions can be drawn from an exhaustive combination of sample size and LFC, broken down by cell type (Additional file 2: Figs. S8-S20). The simulation results implied that the performance of each method is affected by the cell type proportions, i.e., better performance in higher-abundance cell types. All methods are less powerful for the cell subpopulations at lower frequencies ($< 10\%$), and thus the performances are often similar.

Figure 3C shows the heatmap of average AUC, combining all cell types, at different sample size and LFC. This shows the aggregated performance across all cell types in a synthetic replicate, whereas two out of the six cell types do not have any csDEG. Here, ISLET still achieves the highest AUC, at all scenarios. TOAST and TCA's rankings follow immediately after ISLET, in most cases. Overall, in Fig. 3, ISLET demonstrates considerably improved and consistent csDEG detection performance, with the help of improved reference panel recovery.

We also performed simulations to compare the impact of modeling versus the impact of imprecise proportions on csDE testing power. Results are compiled in

Additional file 2: Figs. S21-S22, with sample size at $N = 25, 50, 75, 100$ participants per group. Using estimated proportions would negatively impact all methods, but the order of methods would retain. TDR and ROC curves show the advantage of using ISLET compared with other methods, within either the true or estimated proportions. It is worth noting that ISLET with deconvoluted proportions could outperform other methods with true proportions, at certain top-gene cutoff and towards the tail of ROC curves, despite ISLET maintains a comparable performance in FDR. The impact of imprecise cellular composition on FDR is profound for each compared method. Furthermore, we compare ISLET and DESeq2 at small sample size with $N = 5$ subjects per group. The results in Additional file 2: Figs. S23-S24 show that in a longitudinal study with small sample size, our model still yields higher sensitivity in csDEG calling than the methods borrowing information across genes, such as DESeq2.

In the meantime, we illustrate the performance of slope test based on the framework in Equation (2) by generating synthetic gene expression with differential slope. The simulation design for this scenario is similar to the above, as described in the "Methods". In Additional file 3, Figs. S25-S28 show the false positive rate (FPR) and sensitivity of ISLET slope test at different LFC and sample size, while the FDR for all cell types is shown in Figs. S29. Briefly, ISLET slope test is powerful in detecting cell-type-specific differential change-rate, with robust type I error in each synthetic data and controlled FDR at larger LFC, sample size. This csDEG slope test is not available in the other tools.

### Type I error under the null

We next evaluated these methods in terms of the validity of their *p*-values under the null distribution, where none of the genes are csDEG. Here, under the null hypothesis, the *p*-value distribution should be uniformly distributed between 0 and 1. Our results showed that, among the six methods, TOAST provided the best-calibrated *p*-values for type I error control, immediately followed by ISLET. TCA and csSAM suffered from conservative issues, evident by the inflated density near *p*-values close to 1 (Fig. 4A). Conclusions were jointly supported by the histogram of *p*-value and

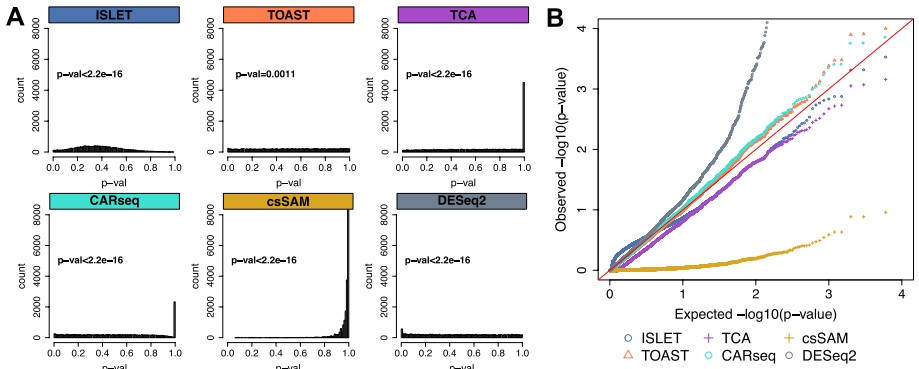

**Fig. 4** Simulation studies for type I error control under the null. **A** Histograms of the observed *p*-values under the null, when no gene is csDEG. The *p*-values shown above histograms are from the Kolmogorov-Smirnov (KS) test under the null hypothesis, for which the contrasted distribution is Uniform [0, 1]. The larger the *p*-value, the more uniform the distribution is. **B** Quantile-quantile plots of the same *p*-values as in **A** but on the $-\log_{10}$ scale. Methods return well-calibrated small *p*-values will stay close to the red diagonal line. $N = 20$ simulations are conducted each setting

by the Kolmogorov-Smirnov (KS) test, which evaluated goodness of fit for *p*-values distributed uniformly. Since the small *p*-values are essential in csDEG calling, we also show the quantile-quantile plot of the *p*-values on -log10 scale for a zoomed-in examination of the small *p*-values (Fig. 4B). CARseq and TOAST have the best-calibrated small *p*-values. ISLET and TCA have roughly calibrated distribution. In contrast, the *p*-values of csSAM and DESeq2 deviate from the expected distribution.

### Application in TEDDY longitudinal bulk transcriptomes

An overview of TEDDY cohort and whole blood bulk RNA-seq data is available in Additional file 4. To apply the competing methods to TEDDY longitudinal bulk RNA-seq data, we selected the participants with IAbs onset between the age of 21 and 27 months and their matched controls. The whole blood samples used in this analysis were collected every 3 months from 9-month age until the first positively-confirmed IAbs onset in the cases (i.e., endpoint), exclusive of the endpoint. Hence, each matched case-control pair had bulk RNA-seq transcriptomes longitudinally profiled at the same age prior to IA seroconversion. Genes with TPM-normalized mean expression less than 1 were removed in downstream analysis. Next, we selected $G = 2000$ genes with top coefficient variation and $< 20\%$ zero counts to perform csDE analysis by each method as illustration. The tools csSAM, CARSeq, TOAST, and DESeq2 were only applicable in the (mean) test described by Eq. (1), while ISLET was capable of detecting csDEG in both mean and change-rate.

We used BH procedure to control FDR in multiple testing, and then reported csDEG at FDR$< 0.1$ in each method. ISLET and TOAST detected DEGs in either B cell or NK cell by the mean test without age-dependent effect. This result was consistent with the findings in [14], i.e., IA-signatures were strongly enriched in B cell and NK cell transcripts, kinases, and transcription factors. For each method, the csDEGs identified in multiple cell populations are shown in Additional file 4: Figs. S30-S34. TCA detected a large number of csDEGs overlapped between different cell types, which may be a result of the inflated false positive rate of TCA. On the other hand, the csDEG called by ISLET or TOAST were not severely overlapped between cell subpopulations, similar to our simulation design. TOAST and DESeq2 identified csDEG partially overlapping with the above ISLET-identified signatures (Figs. S35-S36). For each immune cell type, we performed Gene Sets Enrichment Analysis (GSEA) across 45 candidate REACTOME pathways with size of at least 20 genes, using the rank of test statistics in each method and Bioconductor package fgsea [6, 26]. The significantly enriched pathways were selected by *q*-value$< 0.1$ or *p*-value$< 0.001$ and shown in Fig. 5A. Specifically, ISLET mean test identified potential signature genes differentially expressed in IA cases within NK cells prior to the onset of IAbs. The slope test implemented in ISLET detected the latent differential dynamics in CD4$^+$ T cell (*IGLV1-40*) and NK-cell (*RETN*) preceding IAbs onset, although the relation of CD4$^+$ T cell with IA or T1D was not profound in TEDDY microarray data [14]. These genes showed difference in the change-rate of mixture-cell gene expression (Fig. 5B).

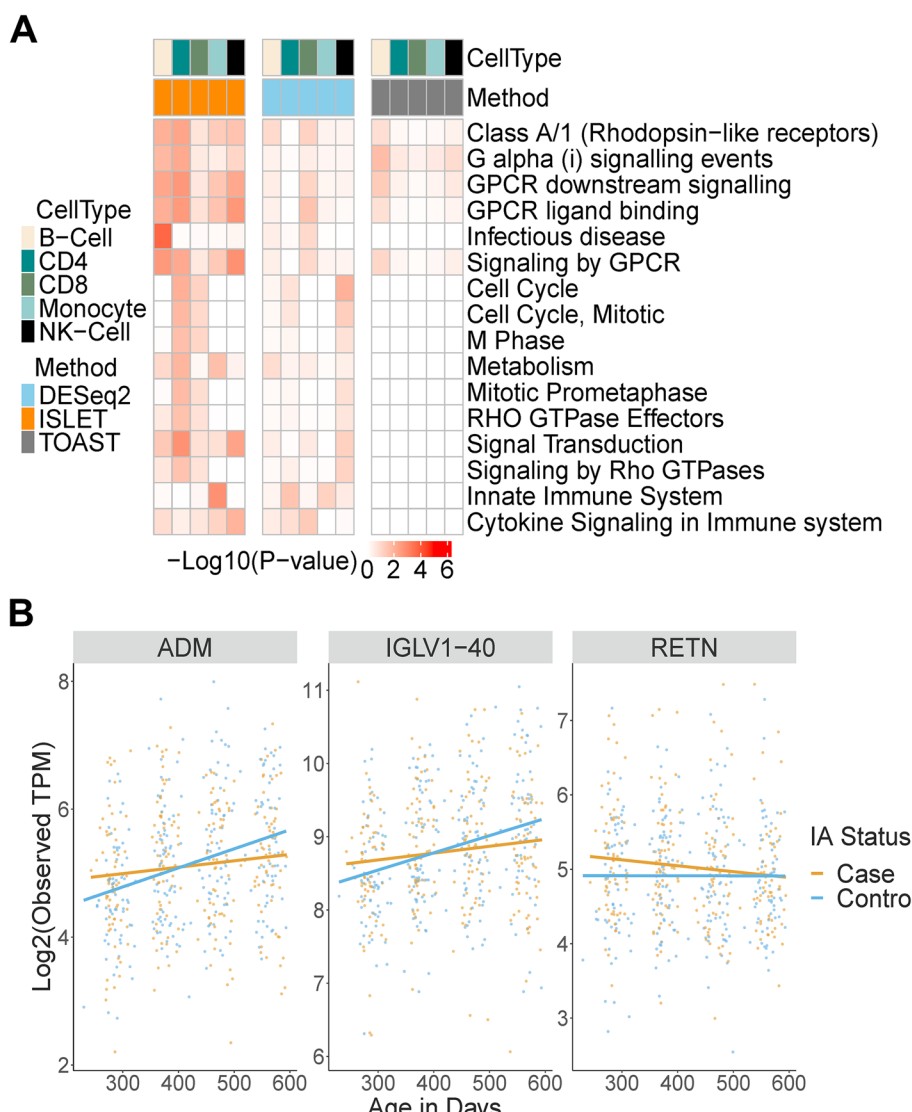

**Fig. 5** Application of ISLET in TEDDY bulk RNA-seq data. **A** Heatmap of *p*-values for pathway enrichment analysis on csDE results from ISLET, DESeq2, and TOAST. **B** Dynamics of csDEG identified by ISLET slope test

**Application in PDBP longitudinal bulk transcriptomes**

We obtained another longitudinal bulk transcriptome dataset from the Parkinson's Disease Biomarker Program (PDBP, https://amp-pd.org/), which is a part of the Accelerating Medicines Partnership (AMP) Parkinson's Disease program. This dataset include both Parkinson's Disease (PD) cases and healthy controls, with repeated measures at baseline, 6, 12, 18, and 24 months. In our analysis, study participants with at least two follow-up visits, besides baseline, were selected. A total of 572 participants (399 PD cases and 173 controls) with 2599 longitudinal blood samples within the two years were included. RNA samples were extracted from whole blood and sequenced. Transcripts per million (TPM) was adopted to represent the gene expression level. We adopted CIBERSORT for deconvolution for cell type proportions. Longitudinal transcriptome data, retrieved cell type proportions, and PD status served

as the inputs for ISLET, TOAST, and DESeq2. Same FDR control procedures were adopted as in the first real data analysis.

As shown in Additional file 4: Section 2, ISLET detected 12 and 6 csDEGs in CD8+ T cells and NK cells, respectively, in the mean test comparing PD and controls. A PD-linked mutation marker *DNAJC6* [27] was called by ISLET as DEG in CD8+ T cells, which was not detected by TOAST or DESeq2. This mutation was also found associated with neurodegeneration in human brain [28]. In addition, the pathway analysis based on ISLET result indicates that the combined csDEGs is enriched in response of EIF2AK4 translation regulation, which is one of the main features of transcriptome profiles in PD [29] and a known pathological alterations in PD and other non-communicable diseases [30]. TOAST called *HLA-DQB2* as DEG in CD8+ T cells, which was found being associated with PD [31]. DESeq2 also detected several genes such as *CCNA2* that had been reported to be associated with PD [32], but these genes were called in multiple cell types by DESeq2 with potential concern of being false positive.

## Discussion

We implemented our model and algorithm in Bioconductor package ISLET to recover reference panel and perform csDEG test. The algorithm implemented in ISLET has several advantages. First, the individual-specific random effect in the reference panel represents the shared information between repeated samples collected from the same participant. This framework also allows the heterogeneity between repeated samples by incorporating age-dependent effect. Second, ISLET does not require the input of single cell gene expression data from a pilot experiment, which is an essential component in some other csDE analysis tools such as PRISM [13]. In addition, the statistical model used in ISLET can be applied to normalized or re-scaled gene expression and the other types of omics data (e.g., microarray gene expression and DNA methylation), because of the Gaussian density employed in EM algorithm. In contrast, the other csDE analysis tools such as CARSeq, PRISM, and DESeq2 are only applicable to RNA-seq counts data.

A limitation in the present research is the lack of longitudinal pure cell transcriptomes from PBMC samples. The temporal cell-type-specific marker genes for IA in the TEDDY cohort cannot be validated in the current bulk RNA-seq data. To benchmark the estimated individual-specific and cell-type-specific gene expression, it is necessary to design an experiment with sorted cells and then compare with the individual reference panel predicted by ISLET. Based on the individual reference profile deconvolved by ISLET, cell type composition can be re-estimated by applying CIBERSORT or the non-negative least square (NNLS) method to the observed longitudinal bulk gene expression and predicted reference panel per subject. This is similar to the option of recalculating cell type proportions in TCA, although TCA constructs the reference panel for each sample. Our future work should iteratively update the cell type proportion estimate and cell-type-specific reference panel.

The current version of ISLET did not incorporate any non-negative restriction in parameter estimation, although the output individual reference panel was truncated at zero and ISLET showed superior performance in reference panel estimation. The parameter estimate in ISLET should be improved by adopting either a restricted EM algorithm with non-negative restrictions [33] or penalized least square regression [34]. Another

improvement in future research is to estimate the cell type proportions and individual-specific gene expression matrix simultaneously. The concurrent deconvolution of longitudinal cell type proportions and individual-specific reference panels may require a complex framework, but this will also reduce the bias in estimated cellular composition and improves the deconvolved individual reference profiles.

## Conclusion

ISLET is a robust and powerful tool for cell-type-specific gene expression recovery and longitudinal differential analysis in bulk RNA-seq data. This framework globally improved the detection of cell-type-specific latent gene signatures.

## Methods

### Overview

Our present work excluded the genes with low expression (mean TPM < 1) or excess zero counts (in > 20% samples) in the real data. The gene-wise overdispersion is a key parameter in the statistical modeling for RNA-seq data, which may vary between solid tissues and whole blood samples. Hence, we used the raw and TPM-normalized counts, individually, to assess the overdispersion pattern in the whole blood bulk RNA-seq data in TEDDY. The scatter plot in Additional file 4: Fig. S39 illustrated that the mean-variance relation in the filtered genes in TEDDY data is similar to that of Poisson or Log-Normal distribution, both can be approximated by Gaussian. In addition, a general Gaussian assumption is favorable in the analysis of other omics data such as DNA methylation and is computationally efficient because of the explicit form of EM estimator. Therefore, we adopted Gaussian density function and the EM algorithm to estimate the group effect and individual-specific random effect.

### Cell-type-specific and individual-specific random effects

To justify our assumption about individual-specific reference matrix and the random effect terms, we employed scRNA-seq and bulk RNA-seq data profiled from blood samples to illustrate the heterogeneity between subjects at cellular level. We first used scRNA-seq PBMC data in [35] to evaluate the inter-subject heterogeneity of cell-type-specific gene expression. The scRNA-seq PBMC raw counts was converted to cell-type-specific counts and normalized by TPM. Genes with mean TPM < 1 were removed. To reduce the potential covariate effect (e.g., disease status) on gene expression, we used control samples in the following analyses. We first employed a chi-squared statistic to confirm that a large proportion of genes in B cells (78%) and CD4+ T cells (62%) had standard deviation greater than half of the empirical mean. Statistical significance of this chi-squared test was determined by FDR < 0.1. In addition, to evaluate the improvement in goodness-of-fit brought by random effects, we compared the gene-wise Akaike information criterion (AIC) of the generalized linear mixed effect model (GLMM) and generalized linear model (GLM), using a paired $t$ test and 1000 most variable genes in TEDDY and PDBP bulk transcriptomic data. We adopted the normal distribution in GLMM and GLM to reduce the impact of density function on AIC. The results showed that GLMM yielded lower AIC than GLM (p< 0.0001) in TEDDY and PDBP data, while

the estimated random effect variance is non-zero for at least 60% and 50% genes per cell type in TEDDY and PDBP data, respectively.

We further applied the function *estimateDisp* in Bioconductor package edgeR to assess intra- and inter-subject heterogeneity in the same scRNA-seq data [35]. The intra-subject overdispersion was estimated based on the longitudinal samples per subject, while the inter-subject overdispersion was evaluated based on the samples at a fixed time point for all subjects. Additional file 5: Figs. S43-S44 show the gene-wise overdispersion between and within subjects by cell type, whereas the inter-subject heterogeneity is significantly larger than intra-subject heterogeneity (Wilcoxon signed rank test $p < 0.0001$). We also evaluated the common overdispersion shared by genes across all time points and at a single time point, individually. The common overdispersion between subjects at a fixed time point was 2.88 for B cells and 1.77 for CD4+ T cells, being close to that of all longitudinal samples, i.e., 3.32 for B cells and 1.92 for CD4+ T cells. Therefore, our current model focuses on the inter-subject variation.

### Mixed-effect regression framework

We use the same model for each gene and implement our algorithm by parallel computing; therefore, we drop the gene index in model description below. Suppose the subject is indexed by $j$, where $j = 1, 2, ..., J$. For each subject $j$, there are $T_j$ longitudinal observations. We use $y_{jt}$ to represent the observed gene expression for subject $j$ at time-point $t$. The dependent variable vector is thus $\boldsymbol{y}$, where $\boldsymbol{y} = (y_{11}, y_{12}, \cdots, y_{1T_1}, \cdots, y_{J1}, y_{J2}, \cdots, y_{JT_J})'$, with length $N = \sum_{j=1}^{J} T_j$. It contains the gene expression values across $J$ subjects' longitudinal observations.

Meanwhile, we also have other inputs that can be treated as known. These include the number of cell types $K$, and the cell type proportions $\theta_{jT_jk}$ for each subject, each sample, and each cell type. Naturally, $\theta_{jT_jk} \in (0, 1)$ and $\sum_{k=1}^{K} \theta_{jT_jk} = 1$. Additionally, we use a binary scalar $z_j$ to indicate the subject's disease status: (e.g., disease $= 1$ vs. normal $= 0$). We then have this following mixed-effect regression model:

$$\boldsymbol{y} = \boldsymbol{X}\boldsymbol{\beta} + \boldsymbol{A}\boldsymbol{u} + \boldsymbol{\varepsilon} \tag{3}$$

where $\boldsymbol{\varepsilon} \sim N(\boldsymbol{0}, \sigma_0^2 \boldsymbol{I})$ are the residuals. Here, $\boldsymbol{X}$ and $\boldsymbol{A}$ are the design matrices for the fixed-effect $\boldsymbol{\beta}$ and random-effect $\boldsymbol{u}$, respectively, where $\boldsymbol{\beta} = (m_1, m_2, \cdots, m_K, \beta_1, \beta_2, \cdots, \beta_K)'$ has two components: $(m_1, m_2, \cdots, m_K)$ are the baseline average gene expression in the control group, and $(\beta_1, \beta_2, \cdots, \beta_K)$ are the difference between the case group and the control group. The random effect $\boldsymbol{u} = (u_{11}, u_{21}, \cdots, u_{J1}, u_{12}, u_{22}, \cdots, u_{J2}, \cdots, u_{1K}, u_{2K}, \cdots, u_{JK})'$ captures the individual-level gene expression deviance from the group-level mean, for each cell type. The design matrices $\boldsymbol{X}$ and $\boldsymbol{A}$ are in the form:

$$
X = \begin{pmatrix}
\theta_{111} & \theta_{112} & \dots & \theta_{11K} & z_1\theta_{111} & z_1\theta_{112} & \dots & z_1\theta_{11K} \\
\theta_{121} & \theta_{122} & \dots & \theta_{12K} & z_1\theta_{121} & z_1\theta_{122} & \dots & z_1\theta_{12K} \\
\dots & \dots & \dots & \dots & \dots & \dots & \dots & \dots \\
\theta_{1T_11} & \theta_{1T_12} & \dots & \theta_{1T_1K} & z_1\theta_{1T_11} & z_1\theta_{1T_12} & \dots & z_1\theta_{1T_1K} \\
\dots & \dots & \dots & \dots & \dots & \dots & \dots & \dots \\
\theta_{J11} & \theta_{J12} & \dots & \theta_{J1K} & z_J\theta_{J11} & z_J\theta_{J12} & \dots & z_J\theta_{J1K} \\
\theta_{J21} & \theta_{J22} & \dots & \theta_{J2K} & z_J\theta_{J21} & z_J\theta_{J22} & \dots & z_J\theta_{J2K} \\
\dots & \dots & \dots & \dots & \dots & \dots & \dots & \dots \\
\theta_{JT_J1} & \theta_{JT_J2} & \dots & \theta_{JT_JK} & z_J\theta_{JT_J1} & z_J\theta_{JT_J2} & \dots & z_J\theta_{JT_JK}
\end{pmatrix}_{N\times 2K}
\tag{4}
$$

$$
A = \begin{pmatrix}
\mathbf{a}_{11} & 0 & 0 & 0 & \dots & \mathbf{a}_{1K} & 0 & 0 & 0 \\
0 & \mathbf{a}_{21} & 0 & 0 & \dots & 0 & \mathbf{a}_{2K} & 0 & 0 \\
0 & 0 & \ddots & 0 & \dots & 0 & 0 & \ddots & 0 \\
0 & 0 & 0 & \mathbf{a}_{J1} & \dots & 0 & 0 & 0 & \mathbf{a}_{JK}
\end{pmatrix}_{N\times Q}
\tag{5}
$$

where $\mathbf{a}_{jk} := (\theta_{j_1k}, \theta_{j_2k}, \cdots, \theta_{jT_jk})'$ is simply a reorganized vector of cell type proportions, to align with random effect $\boldsymbol{u}$.

**Parameters estimation by expectation-maximization algorithm**

The parameter estimation for the Equation (3) can be achieved by an expectation-maximization (EM) algorithm, although other viable approaches exist. Here, to facilitate the setup of the EM algorithm, we first define the "observed" and the "missing" data: $\boldsymbol{w} = (\boldsymbol{y}, \boldsymbol{u}) := (\boldsymbol{w}_{obs}, \boldsymbol{w}_{mis})$, where $\boldsymbol{w}_{obs} := \boldsymbol{y}$ is the observed data of admixed gene expression, and $\boldsymbol{w}_{mis} := \boldsymbol{u}$ is the missing data of individual-level deviance from the group mean. Then, we have the conditional distribution $\boldsymbol{w}_{obs}|\boldsymbol{w}_{mis} = \boldsymbol{y}|\boldsymbol{u} \sim N(\boldsymbol{X\beta} + \boldsymbol{Au}, \sigma_0^2\boldsymbol{I})$ and marginal distribution $\boldsymbol{w}_{mis} = \boldsymbol{u} \sim N(\boldsymbol{0}, \boldsymbol{\Sigma}_u)$. Here, $\boldsymbol{\Sigma}_u$ is a block-diagonal matrix $\boldsymbol{\Sigma}_u = diag(\sigma_1^2\boldsymbol{I}_J, \sigma_2^2\boldsymbol{I}_J, \cdots, \sigma_K^2\boldsymbol{I}_J)$. By calculating the variance-covariance matrix of $\boldsymbol{w}_{mis}$ and $\boldsymbol{w}_{obs}$, we have the following multivariate normal distribution:

$$
\begin{pmatrix} \boldsymbol{w}_{obs} \\ \boldsymbol{w}_{mis} \end{pmatrix} = N\left[ \begin{pmatrix} \boldsymbol{X\beta} \\ \boldsymbol{0} \end{pmatrix}, \begin{pmatrix} \boldsymbol{A\Sigma}_u\boldsymbol{A}' + \sigma_0^2\boldsymbol{I} & \boldsymbol{A\Sigma}_u \\ \boldsymbol{\Sigma}_u'\boldsymbol{A}' & \boldsymbol{\Sigma}_u \end{pmatrix} \right]
\tag{6}
$$

The EM algorithm calculation will then follow naturally.

E-step:

$$
E[\boldsymbol{u}|\boldsymbol{w}_{obs} = \boldsymbol{y}] = \boldsymbol{\Sigma}_u\boldsymbol{A}'\boldsymbol{V}^{-1}(\boldsymbol{y} - \boldsymbol{X\beta})
$$

$$
E\left[\boldsymbol{s}'\boldsymbol{s}|\boldsymbol{w}_{obs} = \boldsymbol{y}\right] = tr(\boldsymbol{A\Sigma}_p\boldsymbol{A}') + (\boldsymbol{A\mu}_p + \boldsymbol{X\beta} - \boldsymbol{y})'(\boldsymbol{A\mu}_p + \boldsymbol{X\beta} - \boldsymbol{y})
$$

$$
E\left[\boldsymbol{u}_k'\boldsymbol{u}_k|\boldsymbol{w}_{obs} = \boldsymbol{y}\right] = tr(\boldsymbol{\Sigma}_{p_k}) + \boldsymbol{\mu}_{p_k}'\boldsymbol{\mu}_{p_k}
$$

Here, $\boldsymbol{s} = \boldsymbol{Au} + \boldsymbol{X\beta} - \boldsymbol{y}$, $\boldsymbol{V} := \boldsymbol{A\Sigma}_u\boldsymbol{A}' + \sigma_0^2\boldsymbol{I}$, $\boldsymbol{\Sigma}_{p_k}$ is the $k$th diagonal block of matrix $\boldsymbol{\Sigma}_p$, and $\boldsymbol{\mu}_{p_k}$ is the $k$th sub-vector in $\boldsymbol{\mu}_p$.

M-step:

For the $(t+1)^{th}$ iteration given the $t^{th}$ iteration:

$$
\hat{\boldsymbol{\beta}}^{(t+1)} = (\boldsymbol{X}'\boldsymbol{X})^{-1}\boldsymbol{X}'(\boldsymbol{y} - \boldsymbol{A}E_{\eta^{(t)}}(\boldsymbol{u}^{(t)}))
$$

$$\hat{\sigma}_0^{2(t+1)} = \frac{E_{\eta^{(t)}}\left[s's|w_{obs} = y\right]}{N}$$

$$\hat{\sigma}_k^{2(t+1)} = \frac{E_{\eta^{(t)}}\left[u_k'u_k|w_{obs} = y\right]}{J}$$

The E-step and M-step above are repeated until convergence. The details of modeling and algorithm is available in Additional file 5.

### Hypothesis testing

To detect csDEG, we can test each hypothesis $H_0 : \beta_k = 0$, individually, or jointly $H_0 : \beta_1 = ... = \beta_K = 0$, by using the (observed) marginal likelihood function $L(\boldsymbol{\beta}, \boldsymbol{\sigma})$ and the likelihood ratio test (LRT) statistic $\lambda$. That is,

$$
\begin{aligned}
L(\boldsymbol{\beta}, \boldsymbol{\sigma}) &= \ln f(\boldsymbol{y}; \boldsymbol{X}, \boldsymbol{A}, \boldsymbol{\beta}, \boldsymbol{\sigma}) \\
&= -\frac{1}{2}\{N \ln(2\pi) + \ln\left|\boldsymbol{A}\boldsymbol{\Sigma}_u\boldsymbol{A}' + \sigma_0^2\boldsymbol{I}_N\right| \\
&\quad + (\boldsymbol{y} - \boldsymbol{X}\boldsymbol{\beta})'(\boldsymbol{A}\boldsymbol{\Sigma}_u\boldsymbol{A}' + \sigma_0^2\boldsymbol{I}_N)^{-1}(\boldsymbol{y} - \boldsymbol{X}\boldsymbol{\beta})\}
\end{aligned}
\tag{7}
$$

and the test statistics:

$$\Lambda = 2(L(\hat{\boldsymbol{\beta}}, \hat{\boldsymbol{\sigma}}) - L(\tilde{\boldsymbol{\beta}}, \tilde{\boldsymbol{\sigma}})) \tag{8}$$

where $\hat{\boldsymbol{\beta}}, \hat{\boldsymbol{\sigma}}$ are the EM estimate for the full model in (3) and $\tilde{\boldsymbol{\beta}}, \tilde{\boldsymbol{\sigma}}$ are the estimate for the reduced model under null hypothesis $H_0$. The test statistic follows chi-square distribution $\lambda \sim \chi_d^2$, where $d$ is the degree of freedom determined by the number of parameters in hypothesis $H_0$. An alternative approach to test csDEG is to apply existing DE analysis methods to the predicted individual-specific gene expression, although this strategy is not ideal due to the possible bias in reference prediction.

### Data generation process in simulation study

Using the procedure described above, we acquire the Dirichlet distribution parameters for cell population composition in controls (denoted as $\bar{\alpha}_C$) and cases (denoted as $\bar{\alpha}_D$), respectively: $\bar{\alpha}_C = (8.85, 6.49, 5.98, 5.28, 4.22, 3.85)$ and $\bar{\alpha}_D = (1.90, 2.25, 2.10, 5.72, 7.33, 15.37)$. To simulate within-group and within-subject overdispersions, two additional parameters ($\xi_Z$ and $\xi_S$) are introduced in the Dirichlet sampling. We first simulate the mean frequencies of $K$ cell types for each subject $j$ in group $Z$ (cases or controls) by Dirichlet distribution $\bar{\boldsymbol{\theta}}_j \sim Dir(\bar{\boldsymbol{\theta}}_Z^{(0)}, \xi_Z)$, where $\bar{\boldsymbol{\theta}}_j$ is a $K \times 1$ vector. The parameters $\bar{\boldsymbol{\theta}}_Z^{(0)}$ and $\xi_Z$ are the expected frequencies and overdispersion for group $Z$. The cell type composition for subject $j$ at time point $t$ is $\bar{\boldsymbol{\theta}}_{jt} \sim Dir(\bar{\boldsymbol{\theta}}_j, \xi_p)$, where $\xi_p$ is the mean overdispersion estimate for the longitudinal samples per subject and represents the heterogeneity between time points.

For gene $g$, the mean expression and overdispersion per subject $j$ are simulated by $\boldsymbol{M}_{gj} \sim MVN(\bar{\boldsymbol{\mu}}_m, \boldsymbol{\Sigma}_m)$, $\boldsymbol{\Phi}_{gj} \sim MVN(\bar{\boldsymbol{\mu}}_\phi, \boldsymbol{\Sigma}_\phi)$. For each cell type $k$, the true reference expression in subject $j$ is generated by Gamma distribution $\lambda_{gjk} \sim \Gamma(\exp(-\Phi_{gjk}), M_{gjk}\exp(\Phi_{gjk}))$, where $M_{gjk}, \Phi_{gjk}$ are components in vectors $\boldsymbol{M}_{gj}, \boldsymbol{\Phi}_{gj}$. The true mixture-cell gene expression level per sample is $\bar{\lambda}_{gjt} = \boldsymbol{\lambda}_{gj}'\bar{\boldsymbol{\theta}}_{jt}$, where

$\lambda'_{gj} = (\lambda_{gj1}, ..., \lambda_{gjK})$ and $\bar{\lambda}_{gjt}$ still follows Gamma distribution [36]. The observed bulk RNA-seq raw counts for gene g in subject *j* measured at time point *t* is generated from Poisson distribution $Y_{gjt} \sim Pois(\bar{\lambda}_{gjt})$. In the scenario of slope test, the reference panel per subject differs between time points, with age effect fixed by (case or control) group. That is shifting $M_{gjk}$ to $\tilde{M}_{gjkt} = M_{gjk} + \Delta_t$ for $t > 1$, where the value of $\Delta_t$ is group-wise. The cell-type-specific true expression per gene per sample at each time point can be updated to $\tilde{\lambda}_{gjkt} \sim \Gamma(\exp(-\Phi_{gjk}), M_{gjkt} \exp(\Phi_{gjk}))$, and the mixture-cell expression is $\bar{\lambda}_{gjt} = \tilde{\boldsymbol{\lambda}}'_{gjt} \bar{\boldsymbol{\theta}}_{jt}$, where $\tilde{\boldsymbol{\lambda}}'_{gjt} = (\lambda_{gj1t}, ..., \lambda_{gjKt})$.

## Supplementary Information

---

**Additional file 1.** Simulation: reference panel estimation.

**Additional file 2.** Simulation: csDEG test in mean.

**Additional file 3.** Simulation: ISLET slope test.

**Additional file 4.** ISLET applications in real data.

**Additional file 5.** Appendix.

**Additional file 6.** Review history.

---

### Acknowledgements

The TEDDY study is funded by the National Institute of Diabetes and Digestive and Kidney Diseases, National Institute of Allergy and Infectious Diseases, National Institute of Child Health and Human Development, National Institute of Environmental Health Sciences, Centers for Disease Control and Prevention, and JDRF. We thank the TEDDY study data coordinating center at Health Informatics Institute, University of South Florida, for data processing and sharing. The PDBP consortium is supported by the National Institute of Neurological Disorders and Stroke (NINDS) at the National Institutes of Health. A full list of PDBP investigators can be found at https://pdbp.ninds.nih.gov/policy. The PDBP investigators have not participated in reviewing the data analysis or content of the manuscript. Data used in the preparation of this article were obtained from the Accelerating Medicine Partnership (AMP) Parkinson's Disease (AMP PD) Knowledge Platform. We thank Lijun Zhang for discussions on PD-related genes and thank Wen Tang for assistance in real data processing.

### Review history

The review history is available as Additional file 6.

### Peer review information

### Authors' contributions

H.F. and Q.L. developed and implemented the algorithm. Z.L. contributed to the methodology discussion. H.F. conceived the experiments. H.F., G.M., T.L., Y.P., and Q.L. conducted the simulations and/or real data analysis. H.F., G.M., and Q.L. wrote the manuscript. H.P. processed TEDDY RNA-seq data and interpreted real data analysis results. J.K. supervised the TEDDY consortium. All authors reviewed the manuscript.

### Funding

This work was partially supported by the National Institutes of Health [U24DK097771 via the NIDDK Information Network's (dkNET) New Investigator Pilot Program in Bioinformatics (PI: Q.L.) and Cancer Center Support Grant P30CA21765 to Q.L.; R03CA270725 to Z.L.], the American Cancer Society Institutional Research Grant (ACS IRG) [IRG-16-186-21 to H.F.] through Case Comprehensive Cancer Center, and the Corinne L. Dodero Foundation for the Arts and Sciences and the Case Western Reserve University (CWRU) Program for Autism Education and Research to H.F. and the American Lebanese Syrian Associated Charities (ALSAC) to Q.L.

### Availability of data and materials

- ISLET is a Bioconductor package with license GPL-2 available at https://bioconductor.org/packages/ISLET [22]
- The TEDDY RNAseq data have been deposited in NCBI's database of Genotypes and Phenotypes (dbGaP) with the primary accession code phs001442.v3.p2
- The PDBP bulk transcriptome and related clinical data are publicly available on request to AMP-PD (https://amp-pd.org/)
- Simulation based on real data: RNA-seq of human immune pure cell lines (https://www.ncbi.nlm.nih.gov/geo/query/acc.cgi?acc=GSE60424)
- Parameter estimation: PROPER package (https://bioconductor.org/packages/PROPER)
- TOAST (https://bioconductor.org/packages/TOAST)
- TCA (https://cran.r-project.org/web/packages/TCA)
- CARseq (https://github.com/Sun-lab/CARseq)
- DESeq2 (https://bioconductor.org/packages/DESeq2)

- csSAM (https://github.com/shenorrLab/csSAM)
- CIBERSORTx (https://github.com/IOBR/IOBR)

## Declarations

### Ethics approval and consent to participate
Not applicable.

### Consent for publication
All authors have read and approved the submission of this manuscript.

### Competing interests
The authors declare that they have no competing interests.

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

## 

