## [**Additional file 6.** Review history. · Genome Biology]

Review History

First round of review

Reviewer 1

Were you able to assess all statistics in the manuscript, including the appropriateness of statistical tests used? Yes: This is a statistical methodology paper and I have assessed the methods, simulations, and evaluations in my review.

Were you able to directly test the methods? No.

Comments to author:

Summary

This paper introduces a new method - ISLET for (a) identification of individual-specific reference panels for expression from longitudinal RNA-seq data and (b) for identification of cell-type specific differentially expressed genes from the same data. This approach uses previously estimated cell-type proportions and then uses an EM-algorithm based estimation procedure to estimate both the model parameters and the reference panel expression. The methodology seems sound and the approach is an improvement for the important, but relatively less common, case of bulk gene expression data measured over time. The paper has all of the characteristics of a strong methodology paper - including comparison to previous approaches, clear articulation of methods, and a full software implementation in a Bioconductor package. I have several minor, but important, suggestions for the paper and one major confusion.

Major issues

1. Figure 1 and the introduction to the method do not clearly enough specify the inputs, assumed quantities, and estimated parameters for the model. It would be very useful to clearly state that the inputs are the temporal expression and case control status; that the model assumes previously known or estimated cell type proportions; and (c) that the model estimates individual specific reference panels. On my first several reads, I thought the model was both estimating the cell type proportions and the reference panels, for example.

Minor issues/clarifications

1. It would be useful to show data from single cell experiments or pure samples where individuals had different expression reference panels. This would help justify the need for the method and should be straightforward to derive from biological replicates in single cell data for example.

2. It would be useful to understand how the variability in the cell type proportion estimates impacts how well you estimate the reference panels. What if those cell type proportion estimates are incorrect?

Reviewer 2

Were you able to assess all statistics in the manuscript, including the appropriateness of statistical tests used? Yes: The method did not have all the required justifications.

Were you able to directly test the methods? No.

Comments to author:

Reviewer report for “ISLET: individual-specific reference panel recovery improves cell-type specific inference”

With the availability of single cell datasets, deconvolving bulk RNA-seq datasets and identifying cell-type specific differential expression is now becoming the de facto analysis approach. The standard practice pre-estimates cell type proportions for each bulk sample and uses these proportions to detect differentially expressed genes at cell-type-specific resolution (there are many existing methods for this type of analysis).

This paper considers a special design where individual subject’s bulk expression is measure over time. They propose a mixed model approach that takes in subject-specific bulk expression over time and pre-estimated cell type proportions of each subject at each time point. Then a random effect term is introduced for each subject’s cell type specific expression to account for the fact that a subject’s cell type specific expression could be correlated across the time points. This is often a reasonable statistical model; however, there are many major concerns regarding the relevance of this model.

(1) the method never tests for these random effects (e.g., $H_0: \sigma^2_1 = \dots = \sigma^2_K = 0$). While the random effect term is conceptually reasonable, it is not clear that it is justified in practice. In fact, it is probably highly dataset dependent. I would have liked to see whether a model with the random effect term provides a significantly better fit for a large collection of longitudinal bulk RNA-seq datasets.

(2) One can easily imagine settings where heterogeneity across times points for subjects is much larger than heterogeneity among subjects at a single time point. In this case, the random effect term might be better off indexing the time – e.g., maybe u_{tk}). Therefore, entertaining these different models by providing a formal testing framework is important for this method to have impact.

(3) The fact that cell type proportions are estimated is not taken into account. In other words, the design matrix “A” would have measurement error when cell type proportions are estimated. I would have expected that this has a more pronounced effect on power in DE estimation than the proposed random effect adjustment.

(4) While the impact of using true versus estimated deconvolution proportions are compared in terms of reference panel estimation, their impact on downstream analysis of identifying cell type specific DE genes is not studied.

(5) Eqn. (1) has u_{jk} in $E[y_{jt}]$. However, $E[u_{jk}]$ should be zero under this model specification (page 14, u is multivariate normal with mean 0). So the notion that the model is estimating subject specific reference expression is a little misleading.

(6) It appears that the model is fitted separately for each gene. This is probably ok for longitudinal studies with many subjects. However, so studies where the number of patients per group is in the order of 5-10, this will likely lead to loss of power. Authors should specify the design regime where the model can be expected to outperform methods that jointly fit genes and borrow information.

(7) FDR levels are set to different values without justification throughout the study.

(8) Finally, methods robustness should be evaluated at multiple actual datasets. The provided example is somewhat limited.

Reviewer 1:

1. *This paper introduces a new method - ISLET for (a) identification of individual-specific reference panels for expression from longitudinal RNA-seq data and (b) for identification of cell-type specific differentially expressed genes from the same data. This approach uses previously estimated cell-type proportions and then uses an EM-algorithm based estimation procedure to estimate both the model parameters and the reference panel expression. The methodology seems sound and the approach is an improvement for the important, but relatively less common, case of bulk gene expression data measured over time. The paper has all of the characteristics of a strong methodology paper - including comparison to previous approaches, clear articulation of methods, and a full software implementation in a Bioconductor package. I have several minor, but important, suggestions for the paper and one major confusion.*

Response: We thank the reviewer for the complimentary comments about our method and manuscript.

2. *Figure 1 and the introduction to the method do not clearly enough specify the inputs, assumed quantities, and estimated parameters for the model. It would be very useful to clearly state that the inputs are the temporal expression and case control status; that the model assumes previously known or estimated cell type proportions; and (c) that the model estimates individual specific reference panels. On my first several reads, I thought the model was both estimating the cell type proportions and the reference panels, for example.*

Response: Thank you for the great suggestion about Figure 1. We apologize for the unclear description about inputs. We revised both **Figure 1** and main text descriptions on **page 3 lines 77-83** to clarify that the inputs are temporal gene expression, disease status, and pre-estimated cell type proportions in panel (A).

3. *It would be useful to show data from single cell experiments or pure samples where individuals had different expression reference panels. This would help justify the need for the method and should be straightforward to derive from biological replicates in single cell data for example.*

Response: Thank you for the constructive advice to add single or pure cell experiments to justify the need for our method. In the revised manuscript, we added PBMC scRNA-seq data to illustrate the inter-subject heterogeneity at cellular level, as described on **pages 13-14 lines 407-440**. Below, we copied the response to comment #6.

“In the revised manuscript, we first used longitudinal single cell PBMC data to evaluate the inter-subject heterogeneity in cell-type-specific gene expression, as described on **page 13 lines 407-415**. To reduce the potential covariate effect (e.g., disease status) on gene expression, we used the longitudinal scRNA-seq data from

controls. The scRNA-seq PBMC raw counts was aggregated as pure cell pseudo bulk counts and normalized by TPM. Genes with mean TPM<1 were removed and the top two abundant cell types were used in the downstream analyses. We first applied a Chi-Squared test statistic to confirm that a large proportion of genes in B cells and CD4+ T cells had standard deviation greater than half of the empirical mean. Statistical significance of this Chi-Squared test was determined by FDR<0.1. We further applied the function *estimateDisp* in Bioconductor package edgeR to assess heterogeneity between subjects, using the samples at a fixed time point. The boxplots in **Figures S43-S44 in Supplementary File 5** show the estimated gene-specific overdispersion between or within subjects per cell type. The Wilcoxon rank sum test confirms that the estimated inter-subject overdispersions are greater than zero ($p<0.0001$).

In addition, we compared the goodness-of-fit of generalized linear mixed effect model (GLMM) and generalized linear model (GLM), using a paired t test on the gene-wise AIC for the top 1000 variable genes in TEDDY bulk transcriptomic data. We adopted the normal distribution in both models to reduce the impact of density function on AIC, implemented in R functions *lmer* and *glm*. The results showed GLMM yielded lower AIC than GLM ($p<0.0001$), with estimated random effect variance $\sigma_{gk}^2 \neq 0$ for at least 60% genes per cell type. We applied the same analysis to another bulk transcriptomic dataset profiled from blood sample, i.e., Parkinson's Disease Biomarker Program (PDBP). The results also showed that GLMM yielded lower AIC than GLM ($p<0.0001$), with estimated $\sigma_{gk}^2 \neq 0$ for at least 50% genes per cell type, as described on **page 14 lines 420-427**. It's worth to note that the current version of ISLET does not employ *lmer* function in parameter estimation, because *lmer* intends to shrink random effect terms to zero (i.e., $\sigma_{gk}^2 = 0$) and consequently, reduces the power of marginal likelihood ratio test."

4. *It would be useful to understand how the variability in the cell type proportion estimates impacts how well you estimate the reference panels. What if those cell type proportion estimates are incorrect?*

Response: We thank the reviewer for bringing up this important aspect in reference panel estimation. We have now added simulations to illustrate this using TOAST, TCA, and ISLET, under samples sizes of 25, 50, 75, and 100; and LFC of 0 (null), 0.5, 0.75, 1, 1.25, and 1.5. Results are added and shown in **Supplementary File 1**, using a 12-panel **Figure S2**. Side-by-side comparisons using ground truth proportions versus using estimated proportions indicates that imprecise cell type proportions would impact reference panel estimation negatively. This is within our expectation and this conclusion holds for all simulation scenarios. When comparing across different methods, we see that our proposed method ISLET still achieves the lowest NMSE on average and the smallest range, compared with other methods, for each simulation setting. These results, overall, highlight both the unfavorable impact of imprecise proportions to reference panel estimation, and the merits of our modeling approach compared with others.

Reviewer 2:

5. *With the availability of single cell datasets, deconvolving bulk RNA-seq datasets and identifying cell-type specific differential expression is now becoming the de facto analysis approach. The standard practice pre-estimates cell type proportions for each bulk sample and uses these proportions to detect differentially expressed genes at cell-type-specific resolution (there are many existing methods for this type of analysis).*

This paper considers a special design where individual subject's bulk expression is measure over time. They propose a mixed model approach that takes in subject-specific bulk expression over time and pre-estimated cell type proportions of each subject at each time point. Then a random effect term is introduced for each subject's cell type specific expression to account for the fact that a subject's cell type specific expression could be correlated across the time points.

Response: We thank the reviewer for the detailed summary of our method and paper.

6. *The method never tests for these random effects (e.g., $H_0: \sigma^2_1 = \dots = \sigma^2_K = 0$). While the random effect term is conceptually reasonable, it is not clear that it is justified in practice. In fact, it is probably highly dataset dependent. I would have liked to see whether a model with the random effect term provides a significantly better fit for a large collection of longitudinal bulk RNA-seq datasets.*

Response: Thank you for the great suggestion of testing for random effects. In the revised manuscript, we first used longitudinal single cell PBMC data to evaluate the inter-subject heterogeneity in cell-type-specific gene expression, as described on **page 13 lines 407-415**. To reduce the potential covariate effect (e.g., disease status) on gene expression, we used the longitudinal scRNA-seq data from controls. The scRNA-seq PBMC raw counts was aggregated as pure cell pseudo bulk counts and normalized by TPM. Genes with mean TPM < 1 were removed and the top two abundant cell types were used in the downstream analyses. We first applied a Chi-Squared test statistic to confirm that a large proportion of genes in B cells and CD4+ T cells had standard deviation greater than half of the empirical mean. Statistical significance of this Chi-Squared test was determined by FDR < 0.1. We further applied the function *estimateDisp* in Bioconductor package edgeR to assess heterogeneity between subjects, using the samples at a fixed time point. The boxplots in **Figures S43-S44 in Supplementary File 5** show the estimated gene-specific overdispersion between or within subjects per cell type. The Wilcoxon rank sum test confirms that the estimated inter-subject overdispersions are greater than zero ($p < 0.0001$).

In addition, we compared the goodness-of-fit of generalized linear mixed effect model (GLMM) and generalized linear model (GLM), using a paired t test on the gene-wise AIC for the top 1000 variable genes in TEDDY bulk transcriptomic data. We adopted the normal distribution in both models to reduce the impact of density function on AIC, implemented in R functions *lmer* and *glm*. The results showed GLMM yielded lower AIC than GLM ($p < 0.0001$), with estimated random effect variance $\sigma_{gk}^2 \neq 0$ for at least 60% genes per cell type. We applied the same analysis to another bulk transcriptomic dataset profiled from blood sample, i.e., Parkinson's Disease Biomarker Program (PDBP). The results also showed that GLMM yielded lower AIC than GLM ($p < 0.0001$), with estimated $\sigma_{gk}^2 \neq 0$ for at least 50% genes per cell type, as described on **page 14 lines 420-427**. It's worth to note that the current version of ISLET does not employ *lmer* function in parameter estimation, because *lmer* intends to shrink random effect terms to zero (i.e., $\sigma_{gk}^2 = 0$) and consequently, reduces the power of marginal likelihood ratio test.

7. *One can easily imagine settings where heterogeneity across times points for subjects is much larger than heterogeneity among subjects at a single time point. In this case, the random effect term might be better off indexing the time – e.g., maybe u_{tk} . Therefore, entertaining these different models by providing a formal testing framework is important for this method to have impact.*

Response: We thank the reviewer for suggesting a formal testing framework for the heterogeneity across time points. In the current model, we only adopted fixed age effect to account for change rate of cell-type-specific expression. We did not use random terms for intra-subject heterogeneity between time points, since this work was motivated by whole blood bulk RNA-seq data and many single cell experiments for blood samples showed ambiguous intra-subject heterogeneity at cellular resolution. We assessed the intra- and inter-subject heterogeneity by using the same PBMC scRNA-seq data, as stated in response #6. **Figure S43-S44 in Supplementary File 5** show gene-wise overdispersion between and within subjects by cell type. A Wilcoxon Signed rank test confirm that the inter-subject overdispersion was significantly larger than intra-subject overdispersion ($p < 0.0001$) per gene. Furthermore, we evaluated the global overdispersion shared by genes across all time points and at a single time point, individually. The global overdispersion of samples at a fixed time point for all subjects was lower than but close to that across time points. These are added on **page 14 lines 430-440**.

In future work, we will add sample-specific random terms at cellular level to consider the variation between time points in the other tissue types, such as primary and recurrent tumors. Incorporating u_{tk}, u_k for sample-specific and subject-specific information in the reference matrix may require advanced algorithms. Meanwhile, we plan to curate a collection of single cell or pure cell experiments with repeated sampling and employ the above overdispersion evaluation framework to test tissue-specific heterogeneity across time points.

8. *The fact that cell type proportions are estimated is not taken into account. In other words, the design matrix “A” would have measurement error when cell type proportions are estimated. I would have expected that this has a more pronounced effect on power in DE estimation than the proposed random effect adjustment.*

Response: We thank the reviewer for bringing up this important aspect in csDE power analysis. By adding additional simulations, we showed the negative impact of imprecise cell type proportion estimation in reference panel estimation; and observed our modeling’s merit in reference panel estimation compared with other approaches, under the same conditions when using imprecise proportions (see main text **page 7 lines 198-204**, and **Supplementary File 1, section 1.2, Figure S2**). For csDE testing power, we also added simulations to compare the impact of modeling versus the impact of imprecise proportions. Results are compiled in **Supplementary File 2, section 1.5, Figure S21**. We evaluated power/sensitivity from TOAST, DESeq2 and ISLET, with ground true cell type proportion versus estimated cell type proportion, at sample size N=25, 50, 75, and 100 subjects per group, and LFC at 0.5, 0.75, 1, 1.25, and 1.5. The power difference between line types (solid vs dashed) illustrates the impact of imprecise cell type proportions, while the difference between colors (blue vs green vs pink) illustrates the effect of using different models. We can see that within each simulation scenario, the imprecise cell type proportion would reduce the statistical power for each compared method. This is within our expectation. The power gain of modeling, in general, would still outweigh the impact of imprecise cell type proportion. These results present another aspect to illustrate the merit of the proposed modeling, even under perturbed estimations. A summary was added in main text **page 9 lines 249-259**.

9. *While the impact of using true versus estimated deconvolution proportions are compared in terms of reference panel estimation, their impact on downstream analysis of identifying cell type specific DE genes is not studied.*

Response: We have now added additional comparisons of using the true versus deconvoluted proportions on csDE calling. We adopted metrics including precision/TDR, ROC curves, sensitivity vs FDR, and statistical power in comparison. Results are compiled into a 16-panel figure and added in **Supplementary File 2, section 1.6, Figure S22**. Sample size was set at 25, 50, 75, and 100 subjects per group. Using estimated proportions would negatively impact all methods, but the order of methods would retain. Precision/TDR and ROC curves show the advantage of using proper modeling (ISLET) compared with other methods, within either the true deconvolution proportions or the estimated alternative. It is worth noting that ISLET with imprecise proportion could outperform other methods with precise proportion, at certain top-gene cutoff and towards the tail of ROC curves. The impact

on FDR is profound on all methods. Here, ISLET will not outperform others in FDR, where it maintains a comparable performance.

10. Eqn. (1) has u_{jk} in $E[y_{jt}]$. However, $E[u_{jk}]$ should be zero under this model specification (page 14, u is multivariate normal with mean 0). So the notion that the model is estimating subject specific reference expression is a little misleading.

Response: We thank the reviewer for pointing out the notion about random effect. The subject-specific random effect is calculated by the posterior mean of u_{jk} given the observed bulk expression and the parameters estimated in the last iteration of EM algorithm, as shown in the first equation in E-step at the bottom of **page 15**. We changed the notion to 'predicts/recover individual-specific reference matrix' throughout the revised manuscript.

11. It appears that the model is fitted separately for each gene. This is probably ok for longitudinal studies with many subjects. However, in studies where the number of patients per group is in the order of 5-10, this will likely lead to loss of power. Authors should specify the design regime where the model can be expected to outperform methods that jointly fit genes and borrow information.

Response: We appreciate for the reviewer's insightful comment about the impact of sample size on the performance of ISLET and methods jointly fitting genes. The first type of methods that jointly fit genes are those estimating overdispersion across genes such as DESeq2, which assumes constant sampling variance of overdispersion estimator across genes. We added a simulation scenario to compare ISLET and DESeq2 at small sample size with $N=5$ participants per group. The results are described on **page 9 lines 259-263** and **Figures S23-24 in Supplementary File 2**. In the longitudinal studies with small sample size, our model still yields higher sensitivity in csDEG calling than DESeq2, although DESeq2 borrows information across genes in overdispersion estimation.

Another type of methods jointly fitting genes for longitudinal gene expression is multivariate modeling, such as multivariate linear mixed effect model (MVLMM). However, MVLMM is not feasible for gene-wise csDE analysis due to the high dimension of transcriptome and multiple mixed effect coefficients for cell types. In future work, one can explore MANOVA-type framework to globally detect csDEG sets at small sample size, but this approach is not applicable to gene-wise csDE analysis.

12. FDR levels are set to different values without justification throughout the study.

Response: We apologize for the different FDR thresholds used in the previous version. In the revised manuscript, we used $FDR < 0.1$ for both simulation and real data analysis. The figures for the called csDEGs in TEDDY were updated in **Supplementary File 4, section 1**. We updated and shortened the summary of TEDDY data analysis results on **page 10 lines 302-319**.

13. *Finally, methods robustness should be evaluated at multiple actual datasets. The provided example is somewhat limited.*

Response: Thank you for the advice of adding more real datasets to evaluate the robustness of ISLET. In the revised manuscript, we applied our method to another bulk RNA-seq data, i.e., Parkinson's Disease Biomarker Program (PDBP), and summarized the results in main text **pages 11-12 lines 324-350 and Supplementary File 4, section 2**. Our method ISLET successfully identified a Parkinson Disease-linked mutation marker differentially expressed in CD8+ T cells, while the compared methods failed to call this marker in csDE testing. For a robust method of csDE testing, we expect trivial or none overlapping between cell types in csDEGs. As shown in **Figure S40, Supplementary File 4**, the csDEGs called by ISLET were not overlapped between CD8+ T cells and NK cells.

Second round of review

Reviewer 1

I reviewed the responses and the revised manuscript. I believe you have sufficiently addressed all of my concerns with the manuscript.